# Nephrotoxic Effects in Zebrafish after Prolonged Exposure to Aristolochic Acid

**DOI:** 10.3390/toxins12040217

**Published:** 2020-03-30

**Authors:** Xixin Wang, Arianna Giusti, Annelii Ny, Peter A. de Witte

**Affiliations:** Laboratory for Molecular Biodiscovery, Department of Pharmaceutical and Pharmacological Sciences, University of Leuven, O & N II Herestraat 49-Box 824, 3000 Leuven, Belgium; xixin.wang@kuleuven.be (X.W.); arianna.giusti@kuleuven.be (A.G.); annelii.ny@kuleuven.be (A.N.)

**Keywords:** zebrafish, nephrotoxicity, aristolochic acid, fibrosis, chronic kidney disorder

## Abstract

With the aim to explore the possibility to generate a zebrafish model of renal fibrosis, in this study the fibrogenic renal effect of aristolochic acid I (AAI) after immersion was assessed. This compound is highly nephrotoxic able to elicit renal fibrosis after exposure of rats and humans. Our results reveal that larval zebrafish at 15 days dpf (days post-fertilization) exposed for 8 days to 0.5 µM AAI showed clear signs of AKI (acute kidney injury). The damage resulted in the relative loss of the functional glomerular filtration barrier. Conversely, we did not observe any deposition of collagen, nor could we immunodetect α-SMA, a hallmark of myofibroblasts, in the tubules. In addition, no increase in gene expression of fibrogenesis biomarkers after whole animal RNA extraction was found. As zebrafish have a high capability for tissue regeneration possibly impeding fibrogenic processes, we also used a *tert^−/−^* zebrafish line exhibiting telomerase deficiency and impaired tissue homeostasis. AAI-treated *tert^−/−^* larvae displayed an increased sensitivity towards 0.5 µM AAI. Importantly, after AAI treatment a mild collagen deposition could be found in the tubules. The outcome implies that sustained AKI induced by nephrotoxic compounds combined with defective *tert^−/−^* stem cells can produce a fibrotic response.

## 1. Introduction

Chronic kidney disease (CKD) is a major global health problem that is characterized by a slow and progressive loss of kidney function. The endpoint of virtually all progressive renal diseases is fibrosis, characterized histologically by an excessive accumulation and deposition of extracellular matrix (ECM) disrupting the normal histo-architecture of the organ. The molecular and cellular pathogenesis of fibrosis is still not fully elucidated. Evidence shows that secondary to repetitive epithelial and/or endothelial cell injury, myofibroblast activation is a key event underlying the fibrotic process [1].

In order to unravel details of fibrogenesis in CKD, and with the aim to find antifibrotic compounds, several models have been generated. These include TGFb1-induced in vitro fibrosis models [2] and in vivo CKD models such as the TGFb1 overexpressing mice model [3] and the unilateral ureteral obstruction (UUO) model [4]. In addition, also chemically-induced models based on the chronic treatment of rodents with compounds like adenine [5], folic acid [6], and aristolochic acid [7,8] have been used.

Aristolochic acid (AA)-induced nephropathy in humans is characterized by progressive renal interstitial fibrosis leading to ESRD and urothelial malignancy and has been observed after unintentional oral intake of *Aristolochia* species [9]. Originally, the disease was reported in Belgian patients taking slimming pills containing Chinese herbal drugs. New cases continue to appear in some parts of the world as local food sometimes is contaminated by AA and herbs that possibly contain AA are not banned yet from traditional medicine practices [10,11,12,13,14]. 

Of interest, similar results as found in patients were observed in rats after chronic AA treatment. For instance, when rats were s.c. injected daily with 10 mg/kg AA, tubular necrosis associated with lymphocytic infiltrates and tubular atrophy surrounded by interstitial fibrosis was present at day 10 and day 35, respectively [7]. Moreover, C57Bl/6J male mice subjected to daily i.p. administration of AA (3.5 mg/kg) already developed clear renal fibrosis from 5 days onwards [8]. Significantly, resident fibroblast activation seems to play a critical role in the process of the renal fibrosis observed [15]. It has also been found that AA-related DNA adducts and proximal tubule injury in rats appear early after AA treatment that is accompanied by an increased expression of TGFb [15,16].

Zebrafish are small freshwater low vertebrates that show high genetic, physiologic, and pharmacological homology with mammals and that can be maintained easily in laboratory conditions at low cost. They are amenable to a wide range of genetic manipulations, supplying tailor-made solutions to specific research questions. As the cell types present in the zebrafish pronephros (early larval stage) and mesonephros (late larval, juvenile, adult stage) closely resemble those found in higher vertebrates, zebrafish have also been used to study kidney function, drug-induced damage and repair mechanisms [17]. Moreover, as zebrafish larvae and juvenile fish are small in size and hence can be arrayed in 24- or 96-well plates, zebrafish-based disease models have become attractive over the last decade as rapid drug discovery platforms [18].

The objective of this study was to generate and validate a zebrafish model of renal fibrosis after aristolochic acid I (AAI) intoxication, for use as a platform dedicated to renal antifibrotics discovery. Nephrotoxic effects of AA were previously studied in zebrafish, but only at high concentrations and short exposure times using animals at their embryonic (18-31 hpf, hours post-fertilization) or early larval stage (3 dpf, days post-fertilization) [19,20]. Herein, we aimed to assess the fibrogenic renal effects of AAI after prolonged exposure of late larval fish (from 15 dpf) to low concentrations of the compound. As zebrafish have a high capability for tissue regeneration [21] possibly impeding fibrogenic processes, we also used a *tert^−/−^* zebrafish line exhibiting telomerase deficiency and impaired tissue homeostasis [22].

## 2. Results

### 2.1. Aristolochic Acid I (AAI) Causes Toxic and Lethal Effects in ZF 

To assess general toxic and lethal effects of chronic AAI treatment, wild-type AB larvae were exposed to different concentrations of the compound from 15 dpf onwards. AAI at 5 µM exhibited an acute toxic effect, and induced rapid and high mortality in ZF (Figure 1A). AAI at a tenfold lower concentration (0.5 µM) produced a somewhat less dramatic effect. In this case a clear delay in the onset of lethal effects was observed, but about 60% of the fish still died after an 11 day treatment period. Exposure to 1 µM AAI elicited a survival pattern that fell between the effects of the high (5 µM) and low (0.5 µM) concentrations. All survival curves were significantly different from the one observed in case of VHC-treated animals.

After 8 DOT (days of treatment), fish exposed to 0.5 µM AAI were smaller than the non-treated fish, and also displayed a curved body (Figure 2A,B). Moreover, their caudal fin showed a less prominent bony ray structure as compared to untreated fish (Figure 2A). In general, food was not detectable in the abdominal cavity of AAI-treated fish. 

As immersion of larvae in 0.5 µM AAI at day 15 post-fertilization resulted in a clear toxic effect after 8 DOT without obvious lethality, we used this condition for further experimental work. 

### 2.2. AAI Causes Morphological Kidney Injury

We generated Tg(*nphs2:*mCherry) and Tg(*enpep:*EGFP) transgenic fish and intercrossed them to create double transgenic zebrafish. Whole-mount fluorescence imaging showed that EGFP-expression under control of the *enpep* promotor is specific for the tubules and ducts in both the pronephric (5 dpf) (Figure 3B) and mesonephric (23 dpf) kidney (Figure 3C). Also the thickening and convolution of the anterior and medial tubules could be easily observed in animals at 23 dpf (Figure 3C). EGFP-related fluorescence recorded by the digital color camera is bluish green (especially observed at 23 dpf) which makes it distinct from the green autofluorescence particularly seen in the gut area (Figure 3B,C). Significantly, the latter fluorescence was also present in non-transgenic WT AB larvae of the same age (results not shown). mCherry expression was limited to the anterior glomeruli at 5 dpf and at 23 dpf (Figure 3B,C), as no clear red spots representing mesonephric glomeruli could be found in the latter case in the medial region (Figure 3C).

We then used the double transgenic Tg(*nphs2*:mCherry) xTg(*enpep*:EGFP) zebrafish to assess morphological kidney injury secondary to prolonged AAI treatment (0.5 µM). Eight days after the start of the treatment (at 15 dpf) the area of the proximal convoluted tubules (PCT) was significantly reduced compared to control conditions (Figure 4A,B,D). In contrast, the EGFP-related fluorescence intensity in the PCT zone was not affected, and also the area and fluorescence intensity of the glomeruli were not modified after AAI treatment (Figure 4A,C,D). 

### 2.3. AA Causes Histological Changes in the Kidney

To confirm these results, we histologically examined the damage in the kidney elicited by prolonged AAI treatment. Transverse sections through the anterior nephron-dense region of kidney tissue of VHC-treated animals showed well-organized glomeruli and flat-shaped tubules surrounded by cells with clear cytoplasm and nuclei positioned asymmetrically to the luminal surface (Figure 5A,a). Conversely, AAI-treated larvae exhibited normal glomeruli but enlarged tubuli with condensed cells with normal nuclei positioned centrally (Figure 5B,b). 

Masson’s trichrome staining of these sections show that AAI treatment did not result in ECM accumulation in the proximal convoluted tubules (PCT) (Figure 6A,a,B,b,E). In the tubule compartment of AAI-treated animals we also could not immunodetect α-SMA, a hallmark of myofibroblasts, and considered to be one of the most important upregulated markers in fibrotic kidney (Figure 7A,B).

### 2.4. AAI Does Not Induce Gene Expression of Fibrogenesis Biomarkers

We also performed qPCR to compare the gene expression level of common molecular biomarkers for fibrogenesis in AAI (0.5 µM) and VHC treated animals (Figure 8). Since it was not possible to selectively collect kidney tissue, RNA was extracted from whole larvae. Significantly, *tgfb1a* and *acta2*, both well identified central mediators in fibrosis [23,24], were not upregulated on gene level after AAI treatment. As TGF-β is responsible for the expression of collagen-related genes [25], we also did not see any upregulation of *col1a1a*, *col4a1* and *fn1a*, and even observed a strong downregulation in case of *col1a1a*. In contrast, *vim,* commonly used as a marker of epithelial to mesenchymal transition (EMT) resulting in renal fibrosis [26], as well as *mmp9*, a matrix metalloproteinase that is an inducer of EMT [27,28] were clearly upregulated. 

As acute treatment of zebrafish embryos to high concentrations of AAI (27.5 µM) resulted in a substantial increase (12.6-fold) of *tnfa* [19], also the gene expression of this inflammation marker was determined. However, no increase in *tnfa* levels after prolonged treatment with AAI was detected. 

### 2.5. AAI Causes Functional Damage to the Kidney

We then assessed the effects of AAI treatment (0.5 µM) on kidney function using *l-fabp*:VDBP-GFP transgenic fish. The estimated molecular weight of the VDBP-GFP fused protein is 79.6 kD close to human albumin (67 kDa), and consequently has been used as a tracer for proteinuria [29]. Significantly, the decrease of fluorescence in the eye circle correlates well with the reduction of intravascular VDBP-GFP fusion protein [30]. The data we present here show that the fluorescence intensity in the eye circle displayed a small but significant decrease in the AAI-treated group (Figure 9A,B), demonstrating functional renal damage. 

### 2.6. AAI-Treatment of Tert^−/−^ Zebrafish Larvae 

The results using AB and transgenic zebrafish larvae showed that prolonged AAI treatment induced kidney injury, but surprisingly without any fibrotic reaction. As the zebrafish kidney has a high regeneration potential [31,32], we then hypothesized that a fast replacement of epithelial cells in the proximal tubule prevented the development of fibrotic tissue. Subsequently, we used telomerase-deficient *tert^−/−^* zebrafish larvae that lack the expression of telomerase that is essential for organ homeostasis [22,33,34]. 

To that end, heterozygous *ter^+/−^* mutants were crossed and the homozygous *tert^−/−^* offspring selected and raised to adulthood. The first-generation are phenotypically normal and fertile, but show a reduced median life span (67 weeks) as compared to the heterozygous fish (>110 weeks) [34]. In this study the second-generation *tert^−/−^* incross progeny with critically short telomeres [34] was used. These *tert^−/−^* fish presented fragility and around 90% embryos were malformed, in agreement with a previous study [34]. 

The larvae thus obtained were substantially smaller than their age-matched counterparts and showed a high mortality between 10 and 26 dpf (Figure 2A,B; Figure 1B). Treated with 0.5 µM AAI from 15 dpf on, G2 *tert^−/−^* zebrafish exhibited a decreased survival rate as compared to VHC-treated animals (Figure 1B). However, the size of *tert^−/−^* fish was not significantly influenced by 8 days of AAI treatment. 

VHC-treated G2 *tert^−/−^* zebrafish showed normal glomeruli in combination with tubules demarcated by cells with condensed cytoplasm and enlarged nuclei (Figure 5C,c). In contrast, AAI-treated *tert^−/−^* larvae displayed disorganized glomeruli with rounded tubules and condensed epithelial cells. Also vacuolated cells were present (Figure 5D,d), a cytopathological condition which has no effect on cell death or can even increase the cell survival potential [35].

Of interest, AAI treatment G2 *tert^−/−^* zebrafish resulted in a small but significant increase in ECM deposit in the tubule region (Figure 6C,c,D,d,E). However, no increased levels of α-SMA could be found (Figure 7C,D). Moreover, also fibrosis-specific genes like *tgfb1a* and *acta2* were not upregulated in G2 *tert^−/−^* zebrafish after AAI treatment, and overall a somewhat similar pattern in up- and downregulated genes could be found as observed in WT AB fish after AAI treatment. However, *vim*, a marker of epithelial to mesenchymal transition (EMT) was not upregulated in AAI treated *tert^−/−^* zebrafish (Figure 8).

## 3. Discussion

Several in vivo rodent models of renal fibrosis are available nowadays, and are used to study its mechanistic underpinnings [3,4,7]. However, although these rodent models possess a high construct and predictive validity mirroring closely the human disease, they are unsuited for screening of large chemical libraries due to their high costs and associated labor-intensive procedures. 

As zebrafish-based platforms offer major advantages over rodent models for the discovery of new therapeutics, we set out in this study to determine conditions required to induce renal fibrosis in zebrafish using AAI. This compound is a potent nephrotoxic that results in renal fibrosis after exposure of rats and humans [7,10]. It has been argued that the mesonephros might represent a more complex and relevant kidney model to study the nephrotoxic potential effects of compounds as compared to the pronephros kidney present in early larval life [36]. We were therefore interested in the effect of AAI on the zebrafish mesonephros that starts developing around 12 dpf, and used late larval animals (15 dpf) that were exposed to AAI for up to 11 days. 

Our results clearly show that AAI had a dramatic effect on the survival of these larvae exposed to the compound from 15 dpf onwards. The outcome therefore is in agreement with previous data demonstrating that the compound is toxic for zebrafish embryos and early larvae [19,20]. Interestingly, we also found that up to 8 days of prolonged exposure to a low AAI concentration (0.5 µM) does not exert a lethal effect on the zebrafish. As we were looking for incubation conditions in zebrafish mimicking the chronic AAI treatment of rodents that resulted in renal fibrosis [7,8], we used the latter experimental set-up for further investigations. 

When the double transgenic Tg(*nphs2*:mCherry) xTg(*enpep*:EGFP) larvae were treated to the selected exposure conditions (0.5 µM, 15–23 dpf), the proximal convoluted tubules (PCT) were significantly shrunk in size, whereas the area related to the glomeruli was not affected. Moreover, AAI-treated wild type (WT) larvae had normal glomeruli, in contrast to the tubuli that presented structural abnormalities. This structural impairment of the tubuli after AAI-treatment likely resulted in a relative loss of the functional glomerular filtration barrier allowing large molecules to pass. This was indicated by the loss from the blood of the VDBP-GFP fused protein, a tracer for proteinuria.

This outcome is in agreement with the selective toxicity of AA for proximal tubular epithelial cells as observed in the acute early phase after s.c. administration of AA to rats [16]. This selectivity has been related to the key role these cells play in the secretion or reabsorption of compounds, particular in the case of AA by expressing organic anion transporters (OATs) [37] that are also present in zebrafish tissues [38]. 

Conversely, we did not observe any deposition of collagen, nor could we immunodetect α-SMA, a hallmark of myofibroblasts, in tubules present in sections through the anterior nephron-dense region of kidney tissue. In addition, we did not find any increase in gene expression of fibrogenesis or inflammation biomarkers after whole animal RNA extraction. In contrast, two markers for epithelial to mesenchymal transition (EMT) (i.e., *vim, mmp9*) were clearly upregulated, typically associated with underlying chronic inflammatory processes and possibly resulting in the transformation of epithelial cells to a fully fibroblastic phenotype [39]. 

Overall, our results reveal that an 8-day exposure of late larval zebrafish to 0.5 µM AAI induces AKI (acute kidney injury) but does not result into a clear fibrotic response. Significantly, fish organs including kidneys possess powerful regenerative abilities after sublethal toxic injury as compared to their humans counterparts, even during adulthood [40,41]. This biological trait originates from the mobilization and differentiation of kidney stem/progenitor cells present during lifetime as non-tubular interstitial cells [41,42]. 

As telomerase activity is essential for the proliferation of cells and critical for the unlimited self-renewal of stem cells [43], we then considered whether exposure of *tert^−/−^* zebrafish to AAI would result in a more dramatic outcome as compared to those found with WT AB fish. Importantly, *tert^−/−^* fish that lack the catalytic subunit reverse transcriptase of telomerase (*tert*), have a dramatically impaired tissue regenerative capacity causing cells to accumulate DNA damage and to become senescent [33,44]. For example, an investigation of the mechanism underlying zebrafish heart regeneration showed that preexisting cardiomyocytes, that normally replace damaged cells after cardiac injury were unable to dedifferentiate and proliferate in *tert^−/−^* zebrafish [44]. Furthermore, a low dose of bleomycin in combination with telomerase-deficiency resulted in pulmonary fibrosis in mice, whereas the same dose in WT animals did not [45]. It was concluded that residing stem cells were not able to replace damaged tissue due to telomere dysfunction, leading to abnormal lung healing by fibroblasts and fibrotic scar formation [45].

In the present study the *tert^−/−^* incross progeny (second-generation *tert^−/−^* mutants or G2 *tert^−/−^*) was used for experimental work as the animals feature shorter telomeres as compared to the first generation, resulting in limited cell proliferation ability and early senescence [34]. In agreement with previous results [34], these G2 *tert^−/−^* larvae showed fragility, a reduced body length and a low survival rate. 

Compared to WT zebrafish, AAI-treated *tert^−/−^* larvae displayed an increased sensitivity towards 0.5 µM AAI. Not only was a lethal effect of the compound noted during the first days of treatment, but also a more severe histological nephrotoxic effect could be observed. Importantly, after 8 days of AAI treatment a mild collagen deposition could be found in the tubules of the mutant animals. This outcome implies that sustained AKI induced by AAI treatment in combination with defective *tert^−/−^* stem cells can result in a local fibrotic response, most likely as a result of a dramatically reduced regenerative ability of the tissue. Contrary to these findings we did not find evidence of an increased expression of some important fibrosis-related genes. Clearly, as the gene expression data were obtained from RNA extracted from whole larvae, the results should be interpreted with some caution, as they might not completely reflect the local changes present in the kidney area. However, we also could not immunodetect myofibroblasts in the PCT region of AAI-treated *tert^−/−^* larvae. Significantly, myofibroblasts within this region were found responsible for collagen deposition in AA-treated rats, after resident fibroblast activation [15]. 

In summary, our data indicate that further investigations are needed to provide a more unequivocal evidence of the fibrotic response in *tert^−/−^* incross progeny. Ideally, this would be achieved by using longer AAI-exposure times anticipating a more robust and pronounced response. For instance, when rats were exposed to 10 mg/kg AA for 10 following days, only an acute phase of tubular necrosis could be observed, and another 25 days of treatment were necessary to induce renal fibrosis [7]. Unfortunately, our results also reveal that only a limited time window is available to treat larval zebrafish with AAI without inducing lethality, limiting dramatically the possibility to go beyond the exposure procedure used. Since daily injections in large sets of larval fish are unconceivable, in this study larvae were immersed to AAI for practical reasons. However, in doing so, skin, gills, and the whole intestinal tract are exposed to a highly toxic compound which likely contributed to a considerable death rate observed in a relatively short period of time. 

In conclusion, in this work we show that the limited time slot and overall induced toxicity dramatically limits the feasibility to deploy AAI as a chemical to set-up a renal fibrosis model in zebrafish. However, it can be expected that other renal fibrogenic compounds like adenine or folic acid [5,6] with substantially lower systemic toxicity could offer advantages over AAI, and further experimental work exploring the renal fibrogenic potential of these compounds in *tert^−/−^* but possibly also in wild type (WT) zebrafish is warranted. 

## 4. Materials and Methods

### 4.1. Zebrafish Lines, Husbandry, and Chemicals

The wild-type AB strain was obtained from the Zebrafish International Resource Center (Eugene, OR, USA). Heterozygous *tert*^+/^*^−^* mutant ZF [22] were a generous gift from Dr. MG Ferreira (University of Sheffield, Sheffield, UK). The homozygous *tert^−/−^* offspring was selected and raised to adulthood, and the *tert^−/−^* incross progeny (second-generation *tert^−/−^* mutants or G2 *tert^−/−^*) used for experimental work. Transgenic *l-fabp*:VDBP-GFP zebrafish [29] were a generous gift from Dr. WB Zhou (University of Michigan, Michigan, MI, USA). 

Adult ZF were kept at 28.0 °C, on a 14/10 h light/dark cycle under standard aquaculture conditions, as described previously [46]. Fertilized eggs were collected via natural spawning. Embryos and larvae were kept in mating tanks in fish medium (1.5 mM HEPES, pH 7.2, 17.4 mM NaCl, 0.21 mM KCl, 0.12 mM MgSO_4_, 0.18 mM Ca(NO_3_)_2_, and 0.6 μM methylene blue), and maintained on a 14/10 h light/dark cycle in an incubator at 28.0 °C. Animals were fed once daily from 5 dpf on with SDS-100 (Scientific Fish Food, Technilab-BMI, CD Someren, the Netherlands). 

Experiments were approved by the Ethics Committee of the University of Leuven (number P007/2016, approval date: 4 February 2016) and by the Belgian Federal Department of Public Health, Food Safety, and Environment (approval number: LA1210199). All procedures were carried out following the Declaration of Helsinki and according to the European Community Council directives 86/609/EEC. 

### 4.2. Aristolochic Acid I Treatment

Aristolochic acid I (AAI, Sigma, Schnelldorf, Germany) was dissolved in dimethyl sulfoxide (DMSO spectroscopy grade, Acros Organics) at a concentration of 10 mM and stored at −20 °C. Larvae were immersed in solutions of AAI by adding the compound to the fish medium at different concentrations (final DMSO concentration: 0.25%) for different time periods starting from 15 dpf (days post-fertilization). Control groups were treated with 0.25% DMSO in fish medium (vehicle, VHC). Fish medium (together with the appropriate AAI and DMSO concentrations) was changed for 50% every 24 h.

### 4.3. Generation of Tg(nphs2:mCherry) and Tg(enpep:EGFP) Transgenic Zebrafish Lines

Tg(*nphs2:*mCherry) transgenic fish expressing mCherry in the glomeruli under the podocin promotor and Tg(*enpep:*EGFP) transgenic fish expressing green fluorescent protein in the pronephric duct and tubules under the glutamyl aminopeptidase promotor (*enpep*) were generated as described previously [36,47]. Briefly, for Tg(*nphs2:*mCherry), a 3.5 kb promotor fragment amplified from Tü zebrafish genomic DNA using primer sequences 5′-CGGTCACCGGAAGTTTATAAGTATATGGG-3′; 5′-AAGAATGTCGAGATGTTTCTGTTTCGGTCC-3′ and the coding sequence for the mCherry were cloned into a multisite Gateway expression vector (Invitrogen, Carlsbad, CA, USA) flanked with *Tol2* recognition sites (Figure 3A). For Tg(*enpep:*EGFP) a 2 kb promotor fragment amplified from Tü zebrafish genomic DNA using primer sequences 5′: AT-CTCGAG-CCTGGTGGAAAAGCGAACAAAGAAAAT; 3′: ATGGATCC-AAGTCAGAACACTCTCTCCCTGCGAAC and the coding sequence for the eGFP were cloned into a multisite Gateway expression vector (Invitrogen) flanked with *Tol2* recognition sites (Figure 3A). 

To generate stable transgenic zebrafish lines, 30 pg of the *nphs2:*mCherry plasmid DNA or 30 pg of the *enpep:*EGFP plasmid DNA was co-injected with 50 pg *Tol2* transposase mRNA into the cytoplasm of single cell stage fertilized nacre zebrafish embryos. Injected embryos were screened for eGFP and mCherry expression at 3–5 dpf. Fluorescent positive individuals (F0) were grown to adulthood and out-crossed with nacre zebrafish. At F2 the two transgenic lines were intercrossed to generate double transgenic embryos (Tg(*nphs2*:mCherry) xTg(*enpep*:EGFP)) to obtain fish with green fluorescent pronephric duct and tubules and red glomeruli. 

### 4.4. Morphological Phenotyping and Lethality 

AB strain larvae and homozygous *tert^−/−^* larvae were exposed to 0.5 µM, 1 µM, 5 µM AAI, and 0.5 µM AAI, respectively, or VHC, from 15 dpf to 26 dpf. Dead fish were counted and removed daily. At 23 dpf, i.e., after 8 DOT (days of treatment), surviving fish (in case of AB larvae, only the ones exposed to 0.5 µM) were examined for major dysmorphologies using a stereo microscope (Leica MZ10 F, Wetzlar, Germany) equipped with a digital color camera (Leica DFC310 FX with Leica Application Suite V3.6 software, (V3.6, Leica, Wetzlar, Germany, 2010). Body length was measured from the anterior tip of the snout to the base of the posterior caudal fin using ImageJ software (1.52p, NIH, Bethesda, MD, USA, 2019), as performed before [48]. The data were normalized to the body length of VHC-treated AB larvae.

### 4.5. Whole-Mount Imaging

Double transgenic Tg(*nphs2*:mCherry) xTg(*enpep*:EGFP) larvae were exposed to 0.5 µM AAI or VHC from 15 dpf onwards. After 8 DOT, larvae were anesthetized in tricaine (0.3 mM in phosphate buffer, pH 7.0) and laterally positioned in 0.1% agarose (Ultrapure, Invitrogen) in Milli-Q water, before being further processed. Brightfield and fluorescent whole-mount imaging was performed by stereo microscopy (Leica MZ10 F). Confocal imaging was performed using a Nikon NiE microscope equipped with a Yokogawa CSU-X spinning-disk module with dual camera (Andor iXon3) in combination with a Plan Fluor 10x objective (NA 0.30). To improve the confocal images, NIS-elements was used (5.11). The images were registered in z-direction and corrected for sample movement. Z-stacks were then EDF processed to convert all z-planes in one image and deconvolved (2D deconvolution Richardson-Lucy (grainy—20 iterations)). Analysis of processed confocal images was performed using ImageJ software. Regions of interest (ROIs) defining the glomeruli (Glu) and the proximal convoluted tubules (PCT) were drawn, and the area and fluorescence intensity were measured. The data were normalized to the results obtained with VHC-treated larvae.

### 4.6. Histological Analysis

AB strain larvae and homozygous *tert^−/−^* larvae were exposed to 0.5 µM AAI or VHC from 15 dpf onwards. After 8 DOT, larvae were anesthetized in tricaine solution and fixed for 4 days in 4% PFA at 4 °C. After washes (2 × 30 min) in cold DPBS (Dulbecco’s phosphate buffered saline, Sigma), fish were decalcified in EDTA solution (684 mM EDTA, pH 8.0). Fish were then washed (2 × 20 min) with DEPC solution and dehydrated through graded alcohol. Before paraffin processing, fish were embedded in 1% agarose (Ultrapure, Invitrogen) in TAE buffer (Thermo Fisher scientific, Vilnius, Lithuania). In order to align the fish, agarose blocks were produced with a plastic mold allowing ten fish to be analyzed simultaneously. Subsequently, samples were serially sectioned transversely at 5 µm thickness using a HM325 manual rotary microtome (Thermo Fisher scientific). H&E staining was processed with Varistain^TM^ Gemini ES Automated Slide Stainer (Thermo Fisher scientific) according to laboratory protocols. 

To visualize collagen deposition, Masson’s Trichrome staining (Sigma-Aldrich) was performed according to the manufacturer’s protocol. Images were taken with an Olympus IX83 inverted microscope with cellSens imaging software (Olympus, IX83, Tokyo, Japan). Regions of interest (ROIs) defining the proximal convoluted tubules (PCT) were drawn, and the images were morphometrically analyzed (relative area occupied by blue coloration) using ImageJ, as described before [49]. The data were normalized to the results obtained with VHC-treated larvae.

Following deparaffination in histo-clear (National Diagnostics, HS-200), antigen retrieval was performed in sodium citrate buffer (SCB) for 40 min at 92 °C. Bovine serum albumin (BSA) (1%) and normal goat serum (Abcam, ab7481) (10%) in tris-buffered saline (TBS) were used as blocking agents. The primary antibody was rabbit anti-α-SMA at 1:300 (Gene Tex, GTX100034) used before successfully to immunodetect α-SMA in zebrafish tissue sections [32]. The secondary antibody was anti-rabbit Alexa Fluor 568 (Thermo Fisher scientific, A-21144) used at 1:200 (α-SMA). Tissues were embedded and counterstained using Fluoroshield mounting medium with DAPI (Abcam, ab104139, Cambridge, UK), and the edges of cover slips were sealed with organic mounting medium (Histolab, 00811-EX, Gothenburg, Sweden). Sections were stored and protected from light at 4 °C before photographing. Images were taken with an Olympus IX83 inverted microscope with cellSens imaging software (Olympus, IX83). 

### 4.7. Quantification of Gene Expression

AB strain larvae and homozygous *tert^−/−^* larvae were exposed to 0.5 µM AAI or VHC from 15 dpf onwards. After 8 DOT, total RNA was extracted using two larvae per condition. Extraction was performed using TRIzol reagent (Life Technologies, Carlsbad, CA, USA) according to the manufacturer’s protocol. 600 ng RNA/sample was reverse transcribed with the High-Capacity cDNA Reverse Transcription Kit (Applied Biosystems, Vilnius, Lithuania). qPCR was performed with the SsoAdvanced Universal SYBR Green Supermix (Bio-Rad, Foster, CA, USA) and the house-keeping gene was 18S. Primers for all genes are listed in Table 1. 

### 4.8. Renal Function Assessment

Transgenic *l-fabp*:VDBP-GFP transgenic fish were treated with AAI or VHC as described. After 8 DOT, whole-mount larvae were examined by fluorescence microscopy. The amount of VDBP present in blood was estimated by measuring the fluorescence intensity of the VDBP-GFP protein in the eye circle (retinal vessel plexus) by ImageJ analysis. The data were normalized to the results obtained with VHC-treated larvae.

### 4.9. Statistical Analysis 

Statistical analysis was performed with log-rank (Mantel-Cox) test (survival rate), two-tailed unpaired Student t-test or Mann–Whitney test for data that failed the normality test, as appropriate (relative PCT size, relative fluorescence intensity, relative PST fluorescence intensity, relative glomeruli fluorescence intensity and EFI). Others (relative body length, percentage of collagen fiber area and qPCR) were analyzed by one-way ANOVA followed by Sidak’s multiple comparisons test with Graphpad Prism 8.

## Figures and Tables

**Figure 1 toxins-12-00217-f001:**
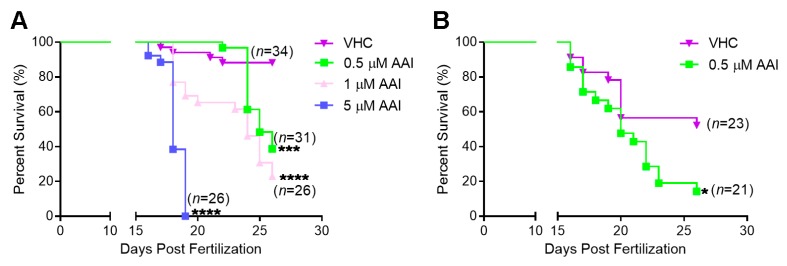
Kaplan–Meier survival plots for AAI-treated AB zebrafish and *tert^−/−^* zebrafish. Larvae were immersed in fish medium complemented with AAI (or VHC) starting from 15 dpf (days post-fertilization). (**A**) survival plots in case of AB fish exposed to 0.5 µM, 1 µM, 5 µM AAI or VHC; and (**B**) in case of *tert^−/−^* fish exposed to 0.5 µM AAI or VHC. Statistical analysis was performed by log-rank (Mantel–Cox) test. *****p* < 0.0001; ****p* < 0.001; **p* < 0.05. Numbers (*n*) shown in the graph refer to the amount of fish present at the start.

**Figure 2 toxins-12-00217-f002:**
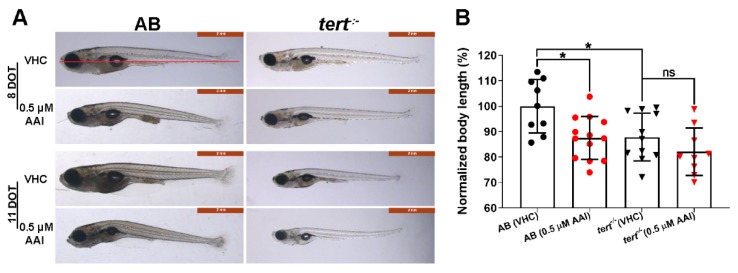
Phenotype and body length of aristolochic acid I (AAI)-treated AB and *tert^−/−^* zebrafish. (**A**) Typical phenotype of AB and *tert^−/−^* larvae after immersion in medium complemented with 0.5 µM AAI (or VHC) starting from 15 dpf for 8 or 11 days (DOT, days of treatment). (**B**) Relative body length of AB (left) and *tert^−/−^* larvae (right) at 8 DOT. The body length was measured with ImageJ from the anterior tip of the snout to the base of the posterior caudal fin (see red line in A) and normalized to VHC-treated AB larvae. Bart charts show the mean ± SD and individual data points (*n* = 9–13 larvae). Statistical analysis was performed by one-way ANOVA followed by Sidak’s multiple comparisons test. **p* < 0.05; ns, no significance.

**Figure 3 toxins-12-00217-f003:**
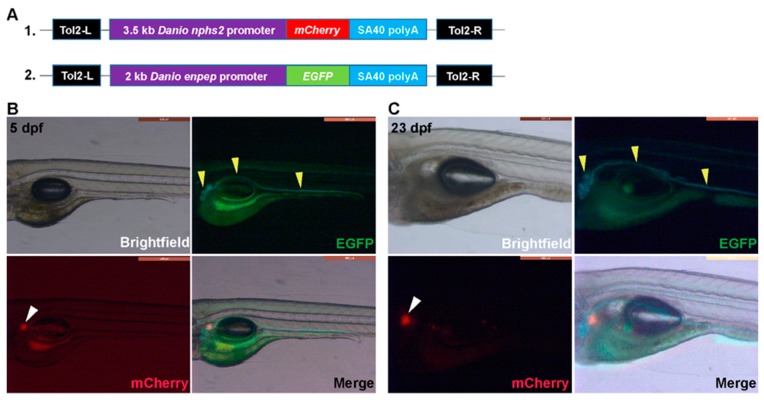
Generation and whole-mount imaging of double transgenic Tg(*nphs2*:mCherry) xTg(*enpep*:EGFP) larvae. (**A**) Schematic graph illustrating the transgene structures of *nphs2*:mCherry and *enpep*:EGFP. Tol2-L and Tol2-R are Tol2 transposon elements to facilitate the transgenesis. SV40 polyA is the polyadenylation signal sequence of simian virus 40; (**B**) and (**C**) Brightfield and fluorescent whole-mount imaging (lateral view, mid segment) of double transgenic Tg(*nphs2*:mCherry)xTg(*enpep*:EGFP) larvae at (**B**) 5 dpf (days post-fertilization) and (**C**) 23 dpf, respectively. Yellow and white arrowheads indicate tubules (including proximal and distal tubules) and glomeruli, respectively.

**Figure 4 toxins-12-00217-f004:**
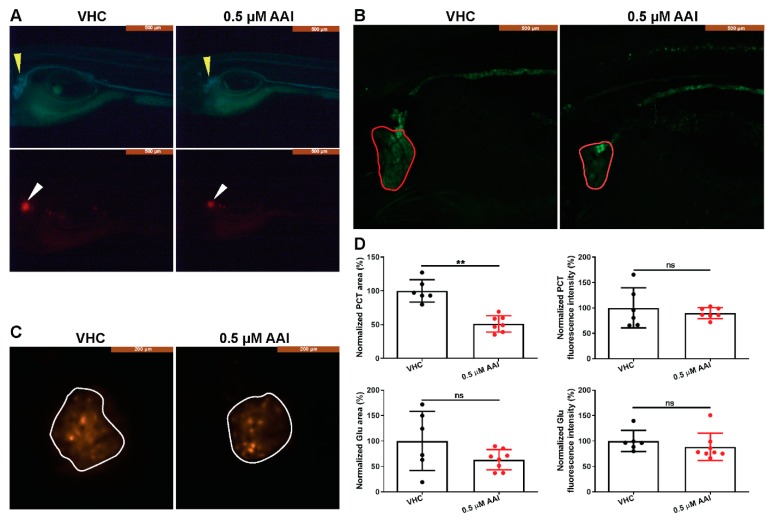
AAI-treated double transgenic Tg(*nphs2*:mCherry) xTg(*enpep*:EGFP) larvae. Larvae were treated with 0.5 µM AAI (or VHC) starting from 15 dpf for 8 days. (**A**) Fluorescent whole-mount imaging (lateral view, mid segment) of the double transgenic larvae. Yellow and white arrowheads indicate tubules and glomeruli, respectively. (**B**) and (**C**) Confocal whole-mount imaging (lateral view) of the (**B**) proximal convoluted tubules (PCT) (outlined in red) and (**C**) glomeruli (outlined in white). (**D**) Quantification of area and fluorescence intensity of the PCT and glomerulus regions by ImageJ. Data were normalized to VHC-treated larvae. Bart charts show the mean ± SD and individual data points (*n* = 6–8 larvae). Two-tailed unpaired Student t-test or Mann–Whitney test for data that failed the normality test, as appropriate (***p* < 0.01; ns, no significance). Glu: glomerulus.

**Figure 5 toxins-12-00217-f005:**
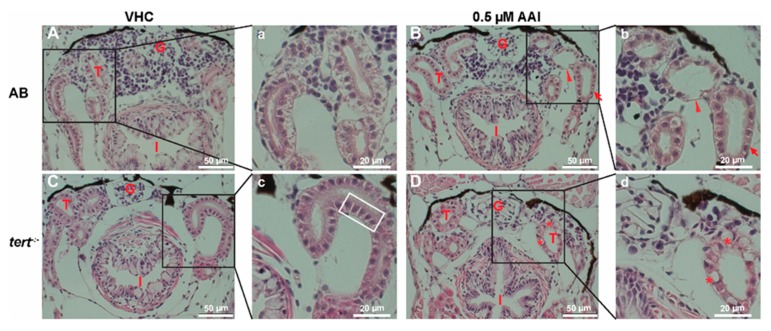
Effect of AAI on kidney histo-architecture (H&E staining). Larvae were treated with 0.5 µM AAI (or VHC) starting from 15 dpf for 8 days. (**A**–**D**) Transverse sections (H&E stained) through the anterior nephron-dense region of kidney tissue of AB (panels **A**, **B**) and *tert^−/−^* larvae (panels **C**, **D**). Panels (a–d) represent magnified parts (black boxes in **A**–**D**). VHC-treated *tert^−/−^* fish had normal glomeruli and tubules presenting cells with condensed cytoplasm and enlarged nuclei (outlined by white rectangle). AAI-treated AB fish showed normal glomeruli and enlarged tubuli (indicated by arrowhead) with condensed cells (arrow), while AAI-treated *tert^−/−^* fish displayed disorganized glomeruli with rounded tubules and vacuolated cells (indicated by asterisks). Numbers of larvae were: *n* = 5 (VHC-treated AB fish), *n* = 5 (AAI-treated AB fish), *n* = 5 (VHC-treated *tert^−/−^* fish) and *n* = 6 (AAI-treated *tert^−/−^* fish). Typical images are shown. G: glomeruli; T: tubules; I: intestine.

**Figure 6 toxins-12-00217-f006:**
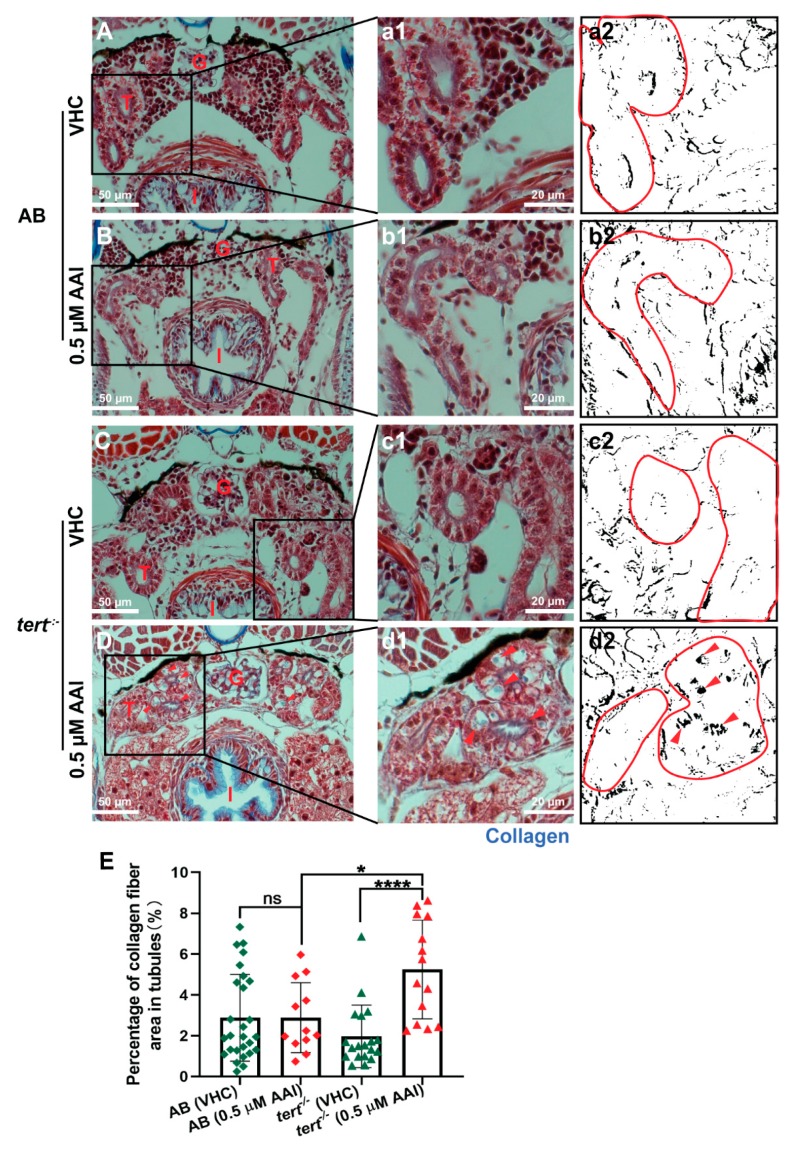
Effect of AAI on collagen accumulation in tubules (Masson’s Trichrome staining). Larvae were treated with 0.5 µM AAI (or VHC) starting from 15 dpf for 8 days. (**A**–**D**) Transverse sections (Masson’s Trichrome stained) through the anterior nephron-dense region of kidney tissue of AB (panels **A**, **B**) and *tert^−/−^* larvae (panels **C**, **D**). Panels (a1–d1) represent magnified parts (black boxes in A-D). Collagen fibers are present as blue zones (arrowheads in d). Blue coloration in images was then separated by Image J. The resulting B and W images (black: blue color present in original picture) (a2–d2) were used to quantify the relative amount of black area in the tubules (outlined in red). (E) Quantification of percentage of collagen fiber area in tubules. Numbers of larvae were: *n* = 6 (VHC-treated AB fish), *n* = 4 (AAI-treated AB fish), *n* = 5 (VHC-treated *tert^−/−^* fish) and *n* = 5 (AAI-treated *tert^−/−^* fish). Different slices per fish through the anterior nephron-dense region of kidney tissue were used to identify tubules. Bart charts show the mean ± SD and individual data points (*n* = 12–27 tubule areas). Statistical analysis was performed by one-way ANOVA followed by Sidak’s multiple comparisons test. *****p* < 0.0001; **p* < 0.05; ns, no significance. G: glomeruli; T: tubules; I: intestine.

**Figure 7 toxins-12-00217-f007:**
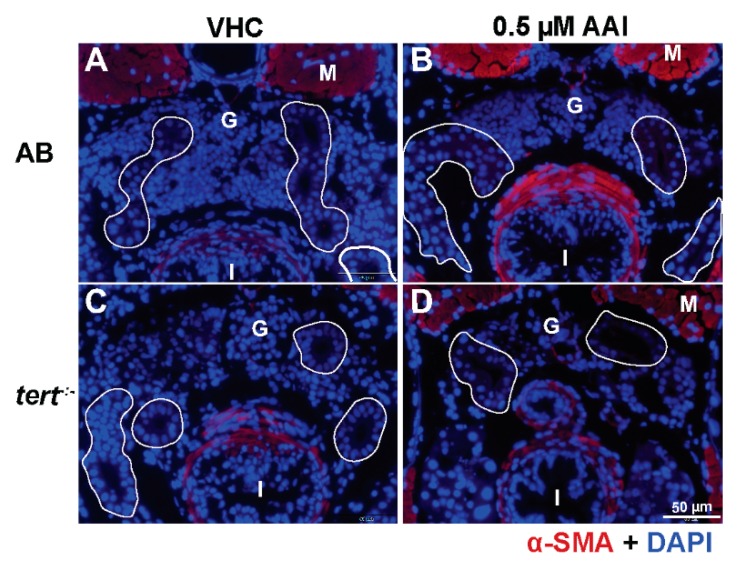
Effect of AAI on α-SMA expression in tubules. Larvae were treated with 0.5 µM AAI (or VHC) starting from 15 dpf for 8 days. (**A**–**D**) Transverse sections through the anterior nephron-dense region of kidney tissue of AB (panels **A**, **B**) and *tert^−/−^* larvae (panels **C**, **D**) after immunostaining for the myofibroblast hallmark α-SMA (red) and DAPI counterstaining (blue). No significant immunostaining for α-SMA is visible within the tubules (outlined in white). Numbers of larvae were: *n* = 5 (VHC-treated AB fish), *n* = 5 (AAI-treated AB fish), *n* = 5 (VHC-treated *tert^−/−^* fish) and *n* = 5 (AAI-treated *tert^−/−^* fish). Typical images are shown. G: glomeruli; M: muscle; I: intestine.

**Figure 8 toxins-12-00217-f008:**
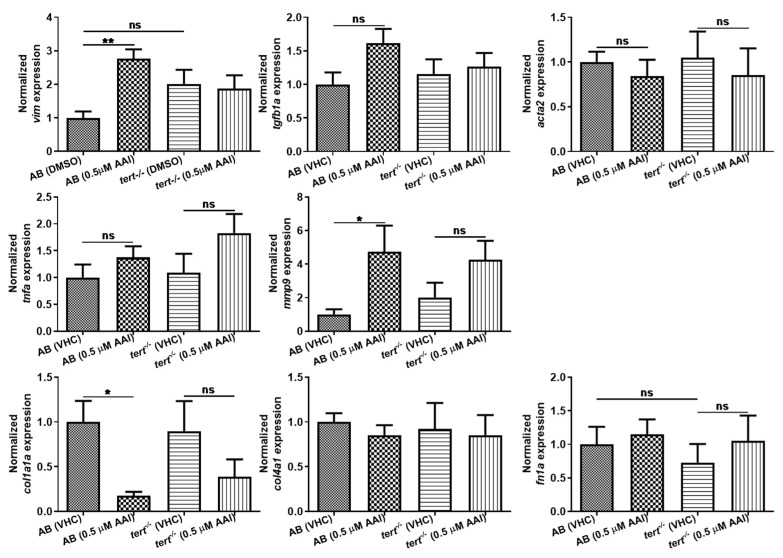
Effect of AAI on gene expression. AB and *tert^−/−^* larvae were treated with 0.5 µM AAI (or VHC) starting from 15 dpf for 8 days. Gene expression level was quantified relative to reference gene 18s ribosomal RNA (18s) expression by ∆∆Cq method. Results for each gene are normalized to VHC-treated AB fish and are presented as mean ± SEM of five independent experiments performed in triplicate. Statistical analysis was performed using one-way ANOVA followed by Sidak’s multiple comparisons test. ***p* < 0.01; **p* < 0.05; ns, no significance.

**Figure 9 toxins-12-00217-f009:**
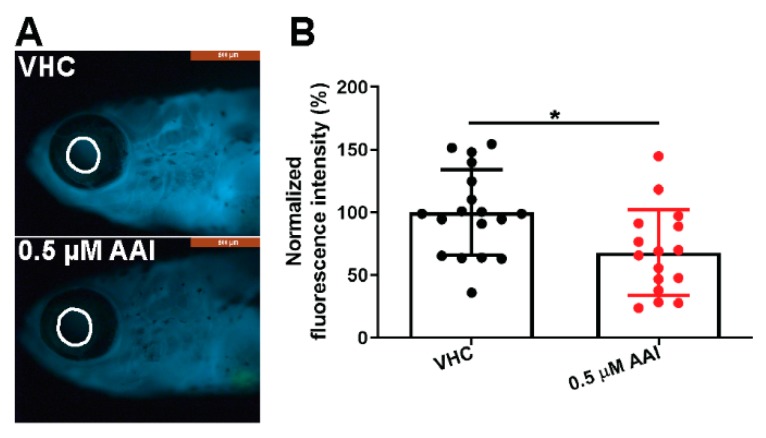
Effect of AAI on eGFR evaluation. *l-fabp*:VDBP-GFP transgenic larvae were treated with 0.5 µM AAI (or VHC) starting from 15 dpf for 8 days. (**A**) Eye fluorescence intensity (EFI) measurements were performed in the area outlined in white; (**B**) Quantification was performed with ImageJ. Bart charts show the mean ± SD and individual data points (*n* = 16–18 zebrafish). Statistical analysis was performed using Mann–Whitney test, two-tailed. **p* < 0.05.

**Table 1 toxins-12-00217-t001:** Gene specific primers used for zebrafish gene expression analysis (Real-time qPCR) in this study.

Gene	Primer Sequence (5’→3’)
*18s*: FW	TCGCTAGTTGGCATCGTTTATG
*18s*: RV	CGGAGGTTCGAAGACGATCA
*acta2*: FW	ACTGTGTAAGGCAGGCTTCG
*acta2*: RV	CACACGGAGCTCGTTGTAGA
*vim*: FW	CCATGGAGGCGTCTGGTTAT
*vim*: RV	TTCCTTCATGGACTCTCGCAG
*tnfa*: FW	GGAGAGTTGCCTTTACCGCT
*tnfa*: RV	CTTGTTGATTGCCCTGGGTCT
*tgfb1a*: FW	CAACGTGTCCGAGATGAAGC
*tgfb1a*: RV	TGGAGACAAAGCGAGTTCCC
*col1a1a*: FW	AGCCCTGGACCTGATGGAAA
*col1a1a*: RV	CACCCTGCTCACCAGACTTT
*col4a1*: FW	GACCACGGCTTCCTTGTAAC
*col4a1*: RV	GTGACCTTTCATTGCCCTGG
*fn1a*: FW	GCAGTGTATGCCGAAAGGAAC
*fn1a*: RV	CAAGTGCAAAGAAGCGTGCT
*mmp9*: FW	ATGGACCTAGAACTGGCCCT
*mmp9*: RV	TGATTTGGCAGGCATCGTCT

FW = Forward; RV = Reverse.

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
