# Peer review of "Nephrotoxic Effects in Zebrafish after Prolonged Exposure to Aristolochic Acid"

_toxins, 2020, doi:10.3390/toxins12040217_

Round 1
Reviewer 1 Report
The manuscript described the evaluation method for aristolochic acid I (AAІ)-mediated nephrotoxic such as fibrosis using zebrafish. The authors demonstrated AAІ toxicities in in vivo assay using transgenic zebrafish. Thus, these findings will be useful for drug screening to discover renal antifibrotics. Therefore, the manuscript is not too excellent to be published. In other words, the manuscript is so excellent that it should be published.
Comments
(1) Can aristolochic acid I elicit toxicities in other tissues different than kidney in zebrafish?
(2) In what mechanisms at molecular level did aristolochic acid I show nephrotoxic due to fibrosis? Was that involved in alfa-SMA, tgfb1a, acta2col1a1a, col4a1, fn1a, or mmp9?
(3) Was aristolochic acid I subject to renal clearance?
(4) Are injured mechanism in kidney based on aristolochic acid I the same between zebrafish and human?
(5) In line of 14, “telomer-ase”, in line of 15, “in-creased”, in line of 16, “depo-sition”, and others, “-“ should be deleted.
That is all.
Reviewer 2 Report
Comment on the article “Nephrotoxic effects in zebrafish after prolonged exposure to aristolochic acid”
The author shows, in this work, the effect in zebrafish after exposure to aristolochic acid (AA) in late larval stage, including the assessment on the survival rate and the morphological changes under different AA concentrations. Furthermore, the histological studies also provide insight on the changes of the gene expression, collagen deposition and alpha-SMA expression. The results were promising to show telomere playing a role in fibrogenic renal effect in zebrafish. Yet, there are a few suggestions to the author prior publication.
First, in the introduction, the author claimed the AA-induced nephropathy is keep appearing in some part of the globe as local food (and water) sometimes is contaminated. However, recent publication supporting this claim is not cited and should be added to the reference list.
Environ Sci Technol. 2020 Feb 4;54(3):1554-1561. doi: 10.1021/acs.est.9b05337.
J Agric Food Chem. 2018 Oct 31;66(43):11468-11476. doi: 10.1021/acs.jafc.8b04770.
J Agric Food Chem. 2016 Jul 27;64(29):5928-34. doi: 10.1021/acs.jafc.6b02203.
Food Chem. 2019 Aug 15;289:673-679. doi: 10.1016/j.foodchem.2019.03.073.
Second, according to the result on collagen deposition, suggested telomerase an essential factor on the collagen deposition, which has been supported by the result in figure 6E. In the discussion, in a lesser extent, was further delineating such observation; therefore, the author shall provide more evidence.
Last, in the summary, the author would like to extend the exposure time to further study the fibrotic response. According to the survival rate study presented in figure 1, there was only 40% and 20% survival rate after 25 days post fertilization in the wild-type and tert -/- zebrafish (i.e. 10 days after 0.5 um AA introduced). The author should take this into account, as longer exposure time would further lower the survival rate and making difficulties on the assessment, which may affect the robustness of the examination.
Apart from these, there are also some minor corrections:
In the abstract, there are a few unnecessary “-“, e.g. telomer-ase, in-creased, etc.
In figure 7, the author should indicate “G” for Glomeruli in the legend.
In figure 8, the author claimed the result were in three independent studies. Yet, the third figure showing the result for tnfa is missing.
Reviewer 3 Report
In this paper the authors tried to generate and validate a zebrafish model of renal fibrosis after aristolochic acid I (AAІ) intoxication. To assess general toxic and lethal effects of chronic AAI treatment, wild-type (WT), AB larvae were exposed to different concentrations of the compound from 15 dpf onwards. AAI had a dramatic effect on the survival of these larvae exposed to the compound from 15 dpf onwards. Overall the results reveal that an 8-day exposure of late larval zebrafish to 0.5 μM AAI induces (acute kidney injury) but does not result into a clear fibrotic response. Compared to WT zebrafish, AAI-treated tert−/− larvae displayed an increased sensitivity towards 0.5 μM AAI
The paper is well written. I found minor misspellings like:
collagen depo-sition
by ne-phrotoxic compounds
The paper can be accepted for publication.
